# Automated monitoring of tweets for early detection of the 2014 Ebola epidemic

**Aditya Joshi**[1]*, **Ross Sparks**[1], **Sarvnaz Karimi**[1], **Sheng-Lun Jason Yan**[2], **Abrar Ahmad Chughtai**[2], **Cecile Paris**[1], **C. Raina MacIntyre**[3,4]

**1** Data61, Commonwealth Scientific and Industrial Research Organisation (CSIRO), Sydney, NSW, Australia, **2** School of Public Health and Community Medicine, University of New South Wales (UNSW), Sydney, NSW, Australia, **3** Kirby Institute, University of New South Wales (UNSW), Sydney, NSW, Australia, **4** College of Public Service & Community Solutions, Arizona State University, Phoenix, AZ, United States of America

* aditya.joshi@csiro.au

**Data Availability Statement:** The dataset available at: https://doi.org/10.25919/5e28cd7a698fb

**Funding:** Raina was supported by a NHMRC Principal Research Fellowship, grant number 1137582. Abrar was employed at and Sheng-Lun

## Abstract

First reported in March 2014, an Ebola epidemic impacted West Africa, most notably Liberia, Guinea and Sierra Leone. We demonstrate the value of social media for automated surveillance of infectious diseases such as the West Africa Ebola epidemic. We experiment with two variations of an existing surveillance architecture: the first aggregates tweets related to different symptoms together, while the second considers tweets about each symptom separately and then aggregates the set of alerts generated by the architecture. Using a dataset of tweets posted from the affected region from 2011 to 2014, we obtain alerts in December 2013, *which is* three months prior to the official announcement of the epidemic. Among the two variations, the second, which produces a restricted but useful set of alerts, can potentially be applied to other infectious disease surveillance and alert systems.

## Introduction

Infectious disease surveillance systems differ in terms of their objective and scope[1]. These systems have traditionally utilised information sources such as health encounters, medical records, hospital statistics, disease registries, over-the-counter drug sales data, laboratory results and surveys. However, for the purpose of early epidemic detection, traditional surveillance data are less timely and sensitive due to factors such as the long process of data validation, influence of bureaucracy, politics, higher costs and resource requirements[2, 3]. The WHO website states that early indicators for more than 60% of epidemics can be found through informal sources such as social media. Therefore, traditional surveillance can be supplemented through the use of publicly available data from internet-based or electronic platforms such as search engines, social media, blogs or forums[4, 5]. When combined with signals and information from traditional sources and agencies, social media-based surveillance of infectious diseases can assist early detection of public health emergencies. While all infections or symptoms may not necessarily get reported on social media due to a variety of reasons, we claim that a change in the prevalence of symptom reports on social media can be an indicator of an outbreak, and can supplement traditional infectious disease surveillance.

was a student at University of New South Wales at the time of the research. Aditya, Cecile, Sarvnaz and Ross were employed by the Commonwealth Scientific and Industrial Research Organisation (CSIRO). The funders had no role in study design, data collection and analysis, decision to publish, or preparation of the manuscript.

**Competing interests:** The authors have declared that no competing interests exist.

In this paper, we investigate if early indicators of the Ebola epidemic in West Africa in 2014 can be obtained from social media posts. This epidemic is regarded as one of the deadliest epidemics in recent times that resulted in a high loss of life and severe stress to medical services in the countries impacted. We evaluate the possibility of surveillance using a dataset of tweets—social media posts published on Twitter—posted between 2011 and 2014 in key cities of the affected region in West Africa. A typical pipeline for a surveillance system to monitor infectious diseases using social media consists of two steps: (a) collection of social media posts reporting symptoms related to the infectious disease; and, (b) application of a monitoring algorithm to generate alerts for an epidemic[6]. For step (a), we employ automatic techniques based on computational linguistics and for step (b), statistical time series monitoring. A lead time of more than two months can be obtained for the Ebola epidemic when these automatic techniques are applied to tweets.

Most past work in social media-based infectious disease surveillance uses datasets created using a set of symptom words. The goal is to signal a disease outbreak before official health systems detect it. This is not to say that social media-based epidemic intelligence would replace human expertise. However, social media brings value in terms of being real-time and originated by human users. However, past work monitors several symptoms of an illness as a collection, and constructs a stream of tweets that is then monitored. For example, for detection of influenza, tweets containing reports of cough, cold, and fever may be considered. However, different symptoms have different prior probabilities of being reported (for example, a fever may be more common than rash or a fever may be reported with less stigma as compared to rash) and different seasonal appearances (for example, cold may be more common in winters while dehydration may be more common in summers). Therefore, tweets reporting each of these symptoms may have different time series behaviours in terms of magnitudes and trends. In this regard, our paper differs from past work in social media-based infectious disease surveillance. Specifically, we address the question:

"*Since different symptoms of a disease may have different attributes, in what ways can tweets that report each of these symptoms be combined to detect the disease*?"

An investigation into this question is likely to impact social media-based syndromic surveillance for other epidemics as well. Towards this, we adapt an architecture that we had previously reported for early detection of disease events using social media[7]. The architecture takes as input a set of tweets and returns a set of alerts, where an alert indicates the possibility of an unexpected behaviour, thereby indicating an epidemic. The original architecture handles only one symptom at a time. To address the question above, we experiment with two variations of the architecture. The first variation uses a combined stream of tweets reporting different symptoms. The second variation obtains alerts from the architecture for each symptom separately and then combines these alerts. We refer to the prior as 'Data Aggregation' because it provides the data for different symptoms together as an input to the architecture, and to the latter as 'Alert Aggregation' because it collects the alerts generated by the architecture for each symptom. To the best of our knowledge, this is the first study that examines whether the monitoring algorithm should handle symptoms separately or together, for social media-based infectious disease surveillance. In addition, it is the first automatic monitoring for the Ebola epidemic using symptomatic data.

## The Ebola epidemic of 2014

The 2014–16 epidemic of Ebola Virus Disease (EVD) in West Africa highlighted the importance of Internet-based surveillance methods. The suspected index case for the epidemic was a

2-year-old boy in Guinea who died on December 6, 2013. The Guinean Ministry of Health first noted an unidentified disease on March 13, 2014. HealthMap retrieved the first public notification on March 14, 2014 from a French news website with headlines reporting a strange fever. Following laboratory confirmation, the WHO released a public statement confirming Ebola on March 23, 2014. Table 1 summarises the timeline of alerts from different sources for the Ebola epidemic of 2014.

The first ProMED report relevant to the Ebola outbreak was a request for information (RFI) on an undiagnosed viral haemorrhagic disease. Hossein et al. [9] found that searching the keyword 'Ebola' in ProMED reports show awareness of spread of Ebola in early April 2014. Similar keyword searches in Google Trends revealed few initial results in March and April, 2014. Alicino et al. [8] compared Ebola-related relative search volumes (RSVs) by regions reported by Google Trends. The highest RSVs were from the three main affected countries. Liberia had the highest score, followed by Sierra Leone and Guinea. However, the most searches for the keyword 'Ebola' occurred on October 16, 2014 when President Obama issued a press release calling National Guard reserves to contain Ebola[8]. In contrast, the first tweet from affected countries mentioning 'Ebola' came as early as December 26, 2013. However, the content of the tweet–"This Ebola of a virus come bad pass HIV . . .May God help us," appeared ambiguous as the user typically posted about football. The next tweet containing Ebola was on March 25th, 2014 –"Guinea has banned the sale and consumption of bats to prevent the spread of the deadly Ebola virus," showing stronger relevance to the outbreak[10]. Different programs or algorithms have been used to extract data from online sources, ranging from text parsing and using the Twitter API to using Google Trends[9, 11, 12]. Some studies did not specify their methods or details relating to data extraction and mining and extent of human moderation[8, 10, 13]. No literature to our knowledge has yet analysed the utility of symptom data from tweets for the detection of EVD[14]. Thus, the aim of our study is to evaluate the value of symptomatic tweets for rapid infectious disease surveillance of the 2014 Ebola epidemic in West Africa, using automated monitoring.

## Architecture

We adapt an architecture that has been reported for early detection of disease events using tweets[7]. This architecture consists of four steps:

1. **Initial selection**: Tweets are selected based on location, date range and keywords. The keywords are words indicating a symptom. The location is derived based on either the tweet location, the author profile location or the location mention. The code is attached as a python notebook, Step 1.ipynb. Please note that Twitter authorisation keys to access the API have been removed since they are personal to this author.

2. **Personal health mention classification**: This step is necessary because a tweet containing a symptom word may not be the report of a symptom. The classifier uses tweet vectors as the

**Table 1. Comparing web-based sources on the West African Ebola epidemic detection and dissemination [8, 9].**

| Source | Date of first detection | Type of dissemination |
|---|---|---|
| Meliandou Health Post | January 24, 2014 | Internal alert |
| Guinea Ministry of Health | March 13, 2014 | Internal alert |
| HealthMap | March 14, 2014 | Textual and graphical alert |
| Bing | March 20, 2014 | No specific information given |
| ProMED-mail | March 22, 2014 | Textual alert, RFI |
| WHO | March 23, 2014 | Textual online statement |

feature representation for a statistical classification algorithm. A tweet vector is the average of word embeddings of content words in a tweet. The word embeddings used are pre-trained on a large corpus. A word embedding is a distributional representation of a word that is expected to capture the semantics of a word. The classification step uses support vector machines trained on a labelled dataset where each tweet is represented as its tweet vector. The code is attached as a python notebook, Step 2.ipynb.

3. **Duplication removal**: We retain the first tweet per day per user. This prevents multiple reports by the same user from swamping the system. It must be noted that this step follows the second step where a classifier has predicted a tweet as a health report. The code is attached as a python notebook, Step 3.ipynb.

4. **Monitoring algorithm**: In this step, we use a monitoring algorithm based on time-between-events[15]. Time-between-events corresponds to the duration between consecutive events in a time series. Of relevance to our algorithm is the event of a tweet being posted. Using in-control data, the algorithm fits a Weibull distribution and estimates its parameters. During test time, the algorithm computes the expected duration between times of posting for consecutive tweets. When the time between consecutive tweets is shorter than an expected value, the tweet is flagged. When p such consecutive tweets are flagged, an alert is generated. The detailed code of the monitoring algorithm has been implemented in R, and is included in the appendix.

In the paper that reported the above architecture[7], we experimented with individual symptoms related to asthma. Therefore, the four steps above were applied in sequence separately for each symptom. Alerts for a symptom were used as a proxy for an alert of the disease. In this paper, we adapt the architecture to be able to handle a collection of symptoms pertaining to the disease being monitored. The two variations are called Data Aggregation and Alert Aggregation.

## Data aggregation

Fig 1 shows the architecture for Data Aggregation. In this case, we use two modules of the initial selection step, one for each symptom. The data is aggregated into a common pool, indicated by the '+'. In other words, tweets for different symptoms are indistinguishable after this stage. Following this, the three steps, namely personal health mention classification, duplication removal and monitoring algorithm, remain the same as in the original architecture. Therefore, the monitoring algorithm works on tweets related to all symptoms of interest together. Since steps 2, 3 and 4 are the same as the base 4-step architecture, data aggregation closely resembles past architecture.

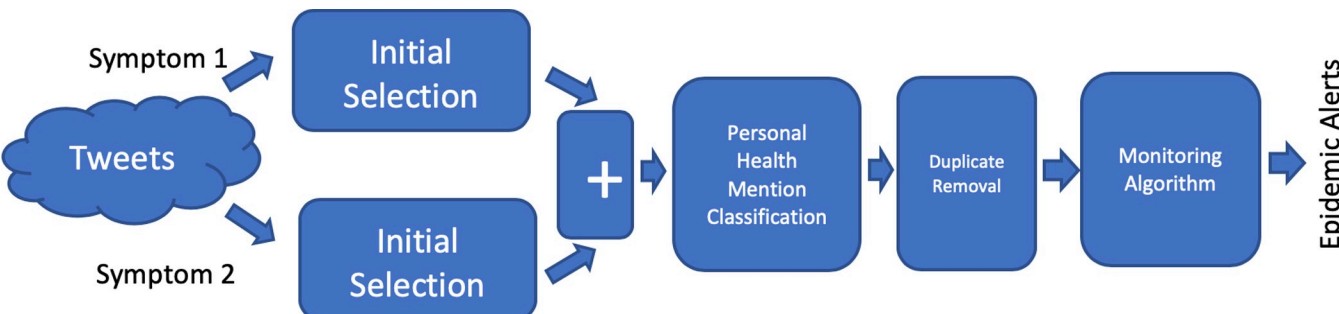

**Fig 1. Adapted architecture using Data Aggregation.**

## Alert aggregation

Fig 2 shows the architecture for Alert Aggregation. In this case, we use multiple channels of the four-step architecture, one channel for each symptom. The final step is a combination where alerts from all channels are combined. This is indicated by the '+'. This combination can be performed in two ways: (a) union, where an alert for a given day is generated by the overall architecture if it was generated by one of the channels; and, (b) intersection, where an alert for a given day is generated by the overall architecture if an alert was generated by all the channels on that day. We refer to these as Alert Aggregation (union) and Alert Aggregation (intersection), respectively.

## Experiment setup

### Data

For our experiments, we created a dataset of tweets using the Twitter Premium Search API (TPSA) (https://developer.twitter.com/en/docs/tweets/search/api-reference/premium-search.html). TPSA allows two key advantages over the free API. It provides full access to historical tweets, while the free API restricts the caller to tweets posted within the last 30 days. TPSA also allows composite search queries that combine location, keyword and date range parameters, while the free API allows calls restricted to one of the parameters (for example, either location or keyword). We used TPSA to download tweets using the following parameters:

1. Date range: December 2011 to December 2014.

2. Locations: We search for locations in key locations in Liberia, Guinea and Sierra Leone. The details are provided in the Appendix. TPSA searches for a location using three arguments: latitude, longitude (indicating the location) and a radius (indicating the distance from the location. This can be up to 40 km). We observed that TPSA returns significantly higher number of tweets from Monrovia, the capital city of Liberia. Therefore, Monrovia is the key location of our dataset. This is expected since, among Guinea, Sierra Leone and Liberia, Liberia had the highest access to the Internet (16.5%, as against 1.6% and 1.7%) during the outbreak[10].

3. Symptoms: For viral haemorrhagic fevers, systems of high sensitivity are expected. To achieve high sensitivity, we use one early symptom: fever, and one late symptom of Ebola: rash. (https://www.who.int/news-room/fact-sheets/detail/ebola-virus-disease). We also experimented with **muscle pain** but the corresponding stream of tweets did not fit the distribution as required by the time-between-events algorithm. We experimented with **bleeding** and **red eyes** but the number of tweets obtained were too few. Therefore, as a simplistic

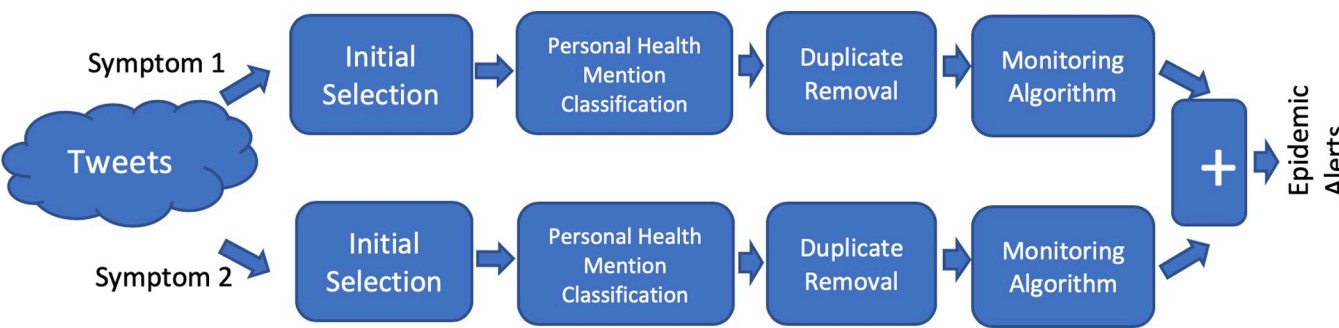

**Fig 2. Adapted architecture for Alert Aggregation.**

setting, we stick to two symptoms: fever and rash. This also makes the union and intersection operations simpler. While the architecture could be applied to more than two symptoms in principle, we restrict to only two symptoms of Ebola.

As a limitation of the API, tweets from those accounts that have been made private or deleted at the time of data collection cannot be accessed. Table 2 shows the number of tweets corresponding to each symptom for the specific date range and locations. Only 24 tweets are common to both the datasets, containing both fever and rash.

A common reluctance against using social media for surveillance stems from questions over its popularity in the affected region. The mobile subscriber penetration rate in 2016 was 43% in Sub-Saharan Africa, lower than the global average of 66% [16]. Only 1.6% of people in Guinea, 1.7% in Sierra Leone and 16.5% in Liberia had access to the Internet during the outbreak[10]. Hence, before we use the automated techniques, we wish to ascertain if there are enough tweets from the locations of interest. Fig 3 shows the daily counts in the aggregated dataset over the date range. We observe spikes in the counts around the period of the epidemic. This result encourages us to apply automated techniques for the Ebola epidemic. The counts do not track the epidemic curve because, when the epidemic becomes known, corresponding chatter is also observed on social media. It is not possible to break down over multiple locations because the tweets returned may not have the exact city name as a text in any of the fields but contain latitude/longitude parameters as present in the geolocation that is tagged with the tweet. We reiterate that, in the rest of the paper, we use a combination of computational linguistics and time series monitoring, and do not rely on manual selection or counting of either tweets or outbreak signals.

## Methods

In our previous work [7], we used two false discovery rates in Step 4, namely the monitoring algorithm: 1 in 1000 and 1 in 2000. The alerts for the false discovery rate of 1 in 2000 were far too few and have not been reported. Similarly, they used two classifiers for the second step, namely the personal health mention classification: SVM and SVM-Perf. We choose support vector machines as the classifier training algorithms because they have shown to perform better than decision trees and random forests for personal health mention classification [17]. For alerts obtained using SVM, we point the reader to the Appendix. The results in the Appendix show that the adapted architecture using SVM-Perf obtains more relevant alerts than the one using SVM. Therefore, in the following section, we report results on SVM-Perf as the classification algorithm and a false discovery rate of 1 in 1000.

## Results

Table 3 shows the alerts generated by the three adapted versions of the architecture from December 2013 to July 2014. In each of these cases, our architecture using social media-based monitoring obtains alerts as early as *December 2013*. This lead time of three months for the Ebola epidemic makes a case for social media-based infectious disease surveillance.

**Table 2. Dataset statistics.**

| Symptom | # tweets |
|---|---|
| Fever | 8507 |
| Rash | 743 |

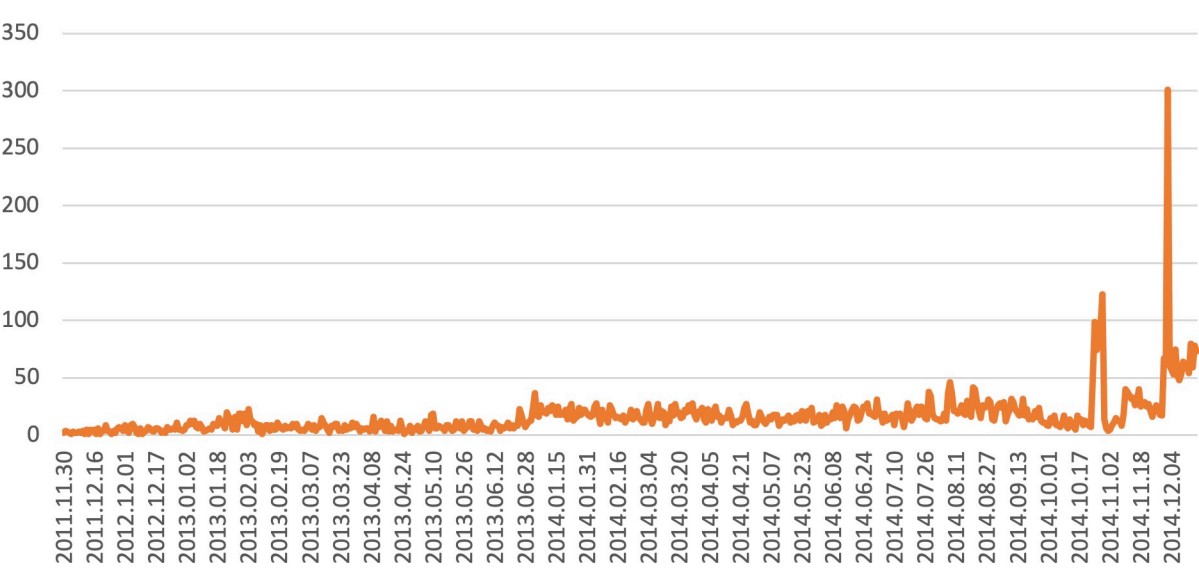

**Fig 3. Daily counts in the aggregated dataset for both the symptoms.**

### Early alerts

As expected, Alert Aggregation (intersection) is restrictive. It only produces one alert: 27th December, 2013. Data Aggregation produces alerts in early December 2013 (i.e., 2nd December onwards), while Alert Aggregation (union) produces alerts in late December 2013 (i.e., 26th December onwards).

### Frequency of alerts

Data Aggregation results in many alerts, starting from early December 2013. In contrast, Alert Aggregation (union) results in few alerts, but they are as early as December 2013 and January 2014. Alerts that are too frequent may not be desirable because they may tend to be ignored.

**Table 3. Alerts generated by the three adapted versions of the architecture.**

|  | Alerts |
|---|---|
| **Data Aggregation** | December 2013: 2, 4, 6, 7, 9, 10, 13, 14, 15, 16, 27, 28, 30 |
|  | January 2014: 3, 4, 6, 10, 11, 13, 17, 18, 20, 24, 25, 27 |
|  | February 2014: 21, 22, 23, 24, 28 |
|  | March 2014: 1 |
|  | April 2014: 24, 25 |
|  | May 2014: 2, 3, 4, 5, 30 |
|  | June 2014: 2, 5, 6, 7, 13, 14, 16, 18, 20, 21, 23, 24, 25 |
|  | July 2014: 11, 12, 14, 15, 16,17, 18, 19, 21, 22, 23, 24, 25, 26, 27, 28, 29 |
| **Alert Aggregation (Union)** | December 2013: 26, 27, 28, 30 |
|  | January 2014: 3, 4 |
|  | February 2014: None |
|  | March 2014: 31 |
|  | April 2014: 1, 27 |
|  | May-July 2014: None |
| **Alert Aggregation (Intersection)** | December 2013: 27 |
|  | January–July 2014: None |

### First alert after the official announcement

For Alert Aggregation (union), we observe an alert on 31[st] March 2014, soon after the official announcement of the epidemic. However, in the case of Data Aggregation, the first alert after the official announcement is on 24[th] April, 2014. Because the data streams are separate in the Alert Aggregation architecture, the alert is sooner than in the case of Data Aggregation.

The complete list of alerts over the entire date range and the alert graphs for the three versions are in the Appendix.

## Conclusions & future work

We adapt an architecture for social media-based infectious disease surveillance and compare two variations: Alert Aggregation and Alert Aggregation. We perform our experiments on a dataset of tweets posted by users in West Africa during the Ebola epidemic of 2014. We focus on fever and rash, two symptoms of Ebola. Our results lead us to two key conclusions:

### Social media provides an alert for the 2014 Ebola epidemic, three months in advance

Using social media-based monitoring, we obtain the earliest alert in December 2013, three months before the announcement. This holds true for all versions of our architecture. It must be noted that the countries of interest have lower internet penetration and lower mobile subscriber penetration than the world average. We show that, despite that, social media-based infectious disease surveillance can lead to early alerts.

### Data aggregation may result in more frequent alerts as compared to alert aggregation

Most work in social media-based surveillance downloads tweets containing a set of keywords and then applies monitoring algorithms (which we refer to as Data Aggregation). We compare this with an approach where tweets related to each symptom are separately analysed using the monitoring algorithm (we refer to this as Alert Aggregation). We observe that Alert Aggregation results in less frequent alerts than Data Aggregation. Therefore, depending on the desired frequency of these alerts, one of the two strategies can be chosen for future work in social media-based infectious disease surveillance.

As future work, our architecture could be adapted for other social media platforms or disease types. Our choice of Twitter as a social media platform is due to the availability of its API for research purposes. The architecture may be applicable to other social media platforms with usage frequency and content similar to Twitter. The architecture may need to be modified for a social media platform that gets used less frequently or has longer posts. Also, we apply the architecture for Ebola in this paper. This could be applied to unexpected diseases such as acute disease events, or common diseases such as influenza.

## Supporting information

**S1 Code.**
(ZIP)

**S1 Appendix.**
(DOCX)

## Author Contributions

**Conceptualization:** Aditya Joshi, Ross Sparks, Sarvnaz Karimi, Abrar Ahmad Chughtai, Cecile Paris, C. Raina MacIntyre.

**Data curation:** Sheng-Lun Jason Yan.

**Formal analysis:** Aditya Joshi, Sarvnaz Karimi.

**Funding acquisition:** Cecile Paris.

**Investigation:** Sarvnaz Karimi.

**Methodology:** Aditya Joshi, Ross Sparks, Sarvnaz Karimi.

**Resources:** Sarvnaz Karimi.

**Software:** Aditya Joshi.

**Supervision:** Cecile Paris, C. Raina MacIntyre.

**Writing – original draft:** Aditya Joshi, Sarvnaz Karimi, Sheng-Lun Jason Yan, Abrar Ahmad Chughtai.

**Writing – review & editing:** Aditya Joshi, Ross Sparks, Sarvnaz Karimi, Cecile Paris, C. Raina MacIntyre.

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
