## [Decision Letter · Decision Letter 0]

5 Dec 2019

PONE-D-19-27142

Automated Monitoring of Tweets for Early Detection of the 2014 Ebola Epidemic

PLOS ONE

Dear Dr. Joshi,

Thank you for submitting your manuscript to PLOS ONE. After careful consideration, we feel that it has merit but does not fully meet PLOS ONE’s publication criteria as it currently stands. Therefore, we invite you to submit a revised version of the manuscript that addresses the points raised during the review process.

Please note that both reviewers have made suggestions to improve the manuscript. You should address all the reviewer concerns including a discussion on how this article's approach improves upon existing, similar approaches.

We would appreciate receiving your revised manuscript by Jan 19 2020 11:59PM. To enhance the reproducibility of your results, we recommend that if applicable you deposit your laboratory protocols in protocols.io, where a protocol can be assigned its own identifier (DOI) such that it can be cited independently in the future. For instructions see: http://journals.plos.org/plosone/s/submission-guidelines#loc-laboratory-protocols

We look forward to receiving your revised manuscript.

Kind regards,

Eric Forgoston

Academic Editor

PLOS ONE

Journal Requirements:

1. Thank you for including your funding statement; "The funders had no role in study design, data collection and analysis, decision to publish, or preparation of the manuscript."

Please provide an amended Funding Statement that declares *all* the funding or sources of support received during this specific study (whether external or internal to your organization) as detailed online in our guide for authors at http://journals.plos.org/plosone/s/submit-now.  

Please state what role the funders took in the study.  If any authors received a salary from any of your funders, please state which authors and which funder. If the funders had no role, please state: "The funders had no role in study design, data collection and analysis, decision to publish, or preparation of the manuscript."

Reviewers' comments:

Reviewer's Responses to Questions

**Comments to the Author**

1. Is the manuscript technically sound, and do the data support the conclusions?

Reviewer #1: Yes

Reviewer #2: Yes

2. Has the statistical analysis been performed appropriately and rigorously? 

Reviewer #1: N/A

Reviewer #2: Yes

3. Have the authors made all data underlying the findings in their manuscript fully available?

Reviewer #1: Yes

Reviewer #2: No

4. Is the manuscript presented in an intelligible fashion and written in standard English?

Reviewer #1: Yes

Reviewer #2: No

5. Review Comments to the Author

Reviewer #1: This is a very interesting article where the investigators used datasets derived from social media (tweets) and applied two adaptations of an existing surveillance architecture with automated monitoring to test for early detection of the 2014 Ebola epidemic. Using this social media-based monitoring, an early alert was detected in December 2013, three months before the official announcement.

The authors took into account the probability and relevance of a symptom, its seasonal appearance and its occurrence over time, which are important in terms of specificity to the disease, when trying to combine symptom reports to detect a disease alert in social media.

Comments:

1. The sentence at the end of the abstract “An additional observation of relevance to infectious disease surveillance in general, is regarding the second adaptation which produces a restricted but useful set of alerts” could be edited in order to convey the message more strongly. It could be rephrased as “The second adaptation, which produces a restricted but useful set of alerts, could potentially be applied in other infectious disease surveillance and alert systems”

2. The authors correctly report that the symptoms they searched for were fever and rash. However, they do not adequately explain why they have not tried to incorporate other (than muscle pain) symptoms of ebola virus disease, like headache, diarrhea, vomiting or bleeding in the architecture.

3. The authors could discuss in more detail if and how their model is applicable in other social media platforms or other (rare or common) diseases.

Reviewer #2: In the paper “Automated monitoring of tweets for early detection of the 2014 Ebola epidemic” the authors present two methods for analyzing the language in social media posts (specifically Twitter “tweets”) to give epidemic alerts. These two methods are described as adaptations of an architecture which has just been accepted for publication elsewhere (the authors citation number 8). Using a data set of tweets from between 2011 and 2014 the authors are able to predict the Ebola epidemic in West Africa three months prior to what they call the official announcement of the epidemic. They do this by analyzing the frequency of the use of words relevant to a particular disease. I find the subject matter of the paper interesting and important, however I do not feel like the authors have done a sufficient job explaining why their approach is different or better from other methods being used in the literature. Although I find it likely that the method presented in the paper improves upon their previous architecture, I would have appreciated it if the need for improvement was addressed. The performance between these two architectures was discussed, but a performance comparison with their previous architecture for this particular data set was not mentioned. Unfortunately the data used is not publicly available, since a premium twitter API requires that the user pay a monthly fee, and so I have to recommend rejection, although I do believe that after some editing the paper deserves to be published, just not in a public library of science journal.

PLOS One has seven criteria for publication which I will address in turn.

1. “The study presents the results of primary scientific research”. The authors present a comparison of two novel architectures on a data set of tweets, and so is a paper describing a new methodology, and does meet PLOS One’s crieteria for “primary scientific research”.

2. “Results reported have not been published elsewhere”. As far as I can tell this is the case.

3. “Experiments, statistics, and other analyses are performed to a high technical standard and are described in sufficient detail”. Other papers that aim to use social media posts to predict disease often have more technical statistical methods for analysis, or are implementing more standard methods of data sorting for prediction, such as Decision Trees and Random Decision Forests. The lack of such methodology is not a failing of the paper, I assume that this is one of the merits of the methodology. If it works about as well, or maybe even better, but is simpler to understand and implement, then it qualifies as preferable by those standards. If that is the case I think that it should be highlighted, and other works should be cited for comparison. I think that the four step method deserves a bit more detail. For instance, “The algorithm computes the expected duration between consecutive tweets. When the time between consecutive tweets is shorter than an expected value, the tweet is flagged”. Although I can imagine how this is being done, I do not find it clearly stated how it is done.

4. “Conclusions are presented in an appropriate fashion and are supported by the data”. The conclusions stated seem to be factual.

5. “The article is presented in an intelligible fashion and is written in standard English”. There are instances in which the grammar needs to be fixed. As I understand it, there is no copy editor, and so the authors must do this themselves. As an example, “Most past work in social media-based infectious disease surveillance uses datasets created using a set of symptom words. before official health systems detect it.”, which appears in the third paragraph of the introduction. This certainly hurts the readability, and for this sentence in particular, does not qualify as intelligible. Other than the overt mistakes, the paper is intelligible.

6. “The research meets all applicable standards for the ethics of experimentation and research integrity”. I see no issues.

7. “The article adheres to appropriate reporting guidelines and community standards for data availability”. It is reported here that “For our experiments, we created a dataset of tweets. We used the Twitter Premium Search API…”. Twitter premium search API has a monthly fee associated with it, and so fails to meet the criteria of PLOS journals “PLOS journals require authors to make all data underlying the findings described in their manuscript fully available without restriction at the time of publication”. Having to pay for the data is a restriction.

6. PLOS authors have the option to publish the peer review history of their article (what does this mean?). If published, this will include your full peer review and any attached files.

Reviewer #1: No

Reviewer #2: No

---

## [Author Response · Author response to Decision Letter 0]

10 Feb 2020

The submission includes a document detailing our responses to the reviewer and editor comments, and corresponding changes made to the manuscript.

---

## [Decision Letter · Decision Letter 1]

27 Feb 2020

Automated Monitoring of Tweets for Early Detection of the 2014 Ebola Epidemic

PONE-D-19-27142R1

Dear Dr. Joshi,

We are pleased to inform you that your manuscript has been judged scientifically suitable for publication and will be formally accepted for publication once it complies with all outstanding technical requirements.

With kind regards,

Eric Forgoston

Academic Editor

PLOS ONE

Reviewers' comments:

Reviewer's Responses to Questions

**Comments to the Author**

1. If the authors have adequately addressed your comments raised in a previous round of review and you feel that this manuscript is now acceptable for publication, you may indicate that here to bypass the “Comments to the Author” section, enter your conflict of interest statement in the “Confidential to Editor” section, and submit your "Accept" recommendation.

Reviewer #1: All comments have been addressed

Reviewer #2: All comments have been addressed

2. Is the manuscript technically sound, and do the data support the conclusions?

Reviewer #1: Yes

Reviewer #2: (No Response)

3. Has the statistical analysis been performed appropriately and rigorously? 

Reviewer #1: Yes

Reviewer #2: (No Response)

4. Have the authors made all data underlying the findings in their manuscript fully available?

Reviewer #1: Yes

Reviewer #2: (No Response)

5. Is the manuscript presented in an intelligible fashion and written in standard English?

Reviewer #1: Yes

Reviewer #2: (No Response)

6. Review Comments to the Author

Reviewer #1: All reviewers' comments have been adequately addressed and significant improvements have been made to the manuscript after suggested revisions and clarifications.

Reviewer #2: (No Response)

7. PLOS authors have the option to publish the peer review history of their article (what does this mean?). If published, this will include your full peer review and any attached files.

Reviewer #1: No

Reviewer #2: No

---

## [Editor Report · Acceptance letter]

3 Mar 2020

PONE-D-19-27142R1 

Automated Monitoring of Tweets for Early Detection of the 2014 Ebola Epidemic 

Dear Dr. Joshi:

I am pleased to inform you that your manuscript has been deemed suitable for publication in PLOS ONE. Congratulations! Your manuscript is now with our production department. 

With kind regards,

on behalf of

Dr. Eric Forgoston 

Academic Editor

PLOS ONE